# COS-OGA Applications in Organic Vineyard Manage Major Airborne Diseases and Maintain Postharvest Quality of Wine Grapes

**DOI:** 10.3390/plants11131763

**Published:** 2022-07-01

**Authors:** Francesca Calderone, Alessandro Vitale, Salvina Panebianco, Monia Federica Lombardo, Gabriella Cirvilleri

**Affiliations:** 1Dipartimento di Agricoltura, Alimentazione e Ambiente, University of Catania, 95123 Catania, Italy; francesca.calderone00@gmail.com (F.C.); monia.lombardo@phd.unict.it (M.F.L.); gcirvil@unict.it (G.C.); 2Dipartimento di Fisica e Astronomia, University of Catania, Via S. Sofia 64, 95123 Catania, Italy; salvina.panebianco@ct.infn.it

**Keywords:** organic vineyards, sustainable management, powdery mildew, gray mold, sour rot, postharvest quality

## Abstract

In most wine-growing countries of the world the interest for organic viticulture and eco-friendly grape production processes increased significantly in the last decade. Organic viticulture is currently dependent on the availability of Cu and S compounds, but their massive use over time has led to negative effects on environment health. Consequently, the purpose of this study was to evaluate the effectiveness of alternative and sustainable treatments against powdery mildew, gray mold and sour rot under the field conditions on Nero d’Avola and Inzolia Sicilian cultivars. In detail, the efficacy of COS-OGA, composed by a complex of oligochitosans and oligopectates, and its effects in combination with arbuscular mycorrhizal fungi (AMF) were evaluated to reduce airborne disease infections of grape. COS-OGA combined with AMF induced a significant reduction in powdery mildew severity both on Nero d’Avola and Inzolia with a mean percentage decrease of about 15% and 33%, respectively. Moreover, COS-OGA alone and combined with AMF gave a good protection against gray mold and sour rot with results similar to the Cu–S complex (performance in disease reduction ranging from 65 to 100%) on tested cultivars. Similarly, the COS-OGA and AMF integration provided good performances in enhancing average yield and did not negatively impact quality and microbial communities of wine grape. Overall, COS-OGA alone and in combination could be proposed as a valid and safer option for the sustainable management of the main grapevine pathogens in organic agroecosystems.

## 1. Introduction

Environmental and food safety issues are driving the wine sector towards innovative systems characterized by eco-friendly and sustainable approaches [1,2]. To this regard, the expansion of the organic viticulture and wine market is increasing more and more and is globally widespread [3]. European countries hold a predominant position in such a scenario, since Spain, France and Italy account for 75% of the world surface destined to organic wine grape production [4]. Italy represents one of the largest organic grape and wine producers, since more than 15% of Italian wine grape cultivation surface is addressed to organic production and covers about 107,143 ha. In detail, Sicily is the first organic viticulture region in Italy with 29,669 ha, corresponding to over 25% of the organic Italian wine-growing surface [4]. Sicilian wine is also well-known for quality and typicality mainly referable to a wide range of indigenous germplasm (i.e., Nero d’Avola, Nerello Mascalese, Nocera, Grillo, Inzolia, Cataratto). European certifications were assigned over time to Sicilian wine production areas, such as 1 “Guaranteed and Controlled Designation of Origin (GCDO)” and 23 “Controlled Designation of Origin (CDOs)”. These labels define the different Sicilian terroirs, in which chemical and fertilization inputs should be reduced to preserve biodiversity and maintain balanced agro-ecosystems [1]. The eco-friendly approach, involving a strong reduction in chemical inputs, is also supported by the agro-food industry and the global government measures. 

Powdery mildew, gray mold and downy mildew, caused by *Erysiphe necator*, *Botrytis cinerea* and *Plasmopara viticola*, respectively, represent major grapevine diseases, that strongly affect yield and quality worldwide. The sour rot, caused by a complex of bacteria and yeast, can be also considered a serious threat for the grape production and winemaking process [5]. The control of these diseases relies almost exclusively on fungicide applications [6,7], including Cu and S compounds which are extensively used in organic vineyards [8,9]. These compounds are considered mandatory steps for disease protection in organic winegrowing due to lack of available alternatives and are often applied simultaneously to control *P. viticola* and *E. necator* infections [10]. Consequently, the massive use of Cu, and to a lesser extent for S compounds, over the last century has led to negative effects for environment health [11,12,13], which are conflicting with the principles of organic agriculture. The European Commission has regulated the use of Cu compounds per year [14] being the maximum allowed 28 kg/ha over 7 years (averagely 4 kg ha^–1^ per year) [15]. Consequently, Cu molecule is considered an active substance candidate for impending withdrawal for agricultural purposes [16] as it has already pursued in some European countries, such as Denmark and the Netherlands [11]. Despite the negative impact, Cu is still necessary in organic viticulture, due to its wide activity spectrum, high efficacy against downy mildew and low cost [8].

According to eco-friendly approaches and increasing organic wine demands, the global scientific community focused on developing alternatives to Cu in order to reduce and/or replace it in main crops, such as grapevine. There are many substances under ongoing testing and validation and some are already on the global market, such as inorganic substances (i.e., zeolite, potassium and sodium hydrogen carbonate); biological control agents (BCAs); biostimulants, including also arbuscular mycorrhizal fungi (AMF) and plant growth-promoting rhizobacteria (PGPR), and resistance inducer (RI) products [11,17,18,19]. 

For example, many BCAs are already commercially available for winegrowers. Several bioformulates based on *Trichoderma* spp., *Streptomyces* spp., *Aureobasidium pullulans*, *Bacillus subtilis* and *B. amyloliquefaciens* are used to manage gray mold infections, whereas *Ampelomyces quisqualis*-based formulates are applied for the control of the powdery mildew of grape [1,17]. With regard to biostimulants, the most promising and widely used are seaweed extracts, protein hydrolysates, humic and fulvic acids, silicon, AMF and PGPR [18]. Several studies showed that mycorrhizal colonization provides pathogen protection through inducing plant systemic resistance [19,20]. AMF are also considered among the viable alternatives for a sustainable vineyard management being able to establish root symbiotic relationship with grapevine such as it happens for some species as *Glomus* and *Rhizophagus* [19,21,22].

Several RIs have been recently studied focusing on their mode of action and performances versus grapevine pathogens [23]. Some of these provided adequate protection in controlled conditions [24,25] and others in vineyards [8,26,27]. Among these, laminarin and chitosan aroused much interest. Romanazzi et al. [27] reported the effectiveness of laminarin and Saccharomyces extracts mixtures and chitosan alone in reducing downy mildew in vineyards although to a lesser extent if compared to copper compounds. Furthermore, chitosan confirmed similar performances against *P. viticola* both under high and low disease pressures [28].

Among the RIs present on the global market, chitosan oligosaccharides (COS)–oligogalacturonides (OGA) complex has been found to induce resistance against biotrophic pathogens in different crop plants [29]. As a consequence of the favorable opinion expressed by the European Food Safety Authority [30], the European Commission [31] approved it as “the first low-risk active substance”. COS-OGA is an active substance resulting from the combination of a complex of COS, which are compounds found in fungal cell walls and crustacean exoskeletons, with pectin-derived OGA originating from plant cell walls. Therefore, chitosan and pectin form a stabilized complex, known as an egg box, which triggers a set of signaling resulting in defense reactions against potential invaders. COS and OGA fragments are detected by plant as non-self and self-molecules, respectively. The coupled danger signals increase the speed and intensity of plant defense response [32]. COS-OGA gave adequate protection against powdery mildew of tomato and cucumber [26,33]. Moreover, COS-OGA showed good activity to prevent potato late blight and root-knot nematode attacks (*Meloidogyne graminicola*) on rice crop [29,34]. Otherwise, little information is available in the literature about COS-OGA performance against grapevine diseases regarding the powdery mildew control in integrated production systems [7,26]. 

Thus, the aim of this paper was the evaluation of the performance of COS-OGA alone and in combination with AMF—commonly used in organic vineyards of this Sicilian district—on Nero d’Avola and Inzolia cultivars (i) in managing powdery mildew, gray mold and sour rot representing key pathogens for Mediterranean grape production; (ii) in reducing yield losses and maintaining postharvest chemical characteristics of grape; and (iii) in affecting the culturable carposphere microflora.

## 2. Results

### 2.1. Field Experiments

The effectiveness of sustainable alternative compounds against powdery mildew, gray mold and sour rot was evaluated through the in-field assessment of symptoms on bunches confirmed by laboratory results. The treatment efficacies were always referred to relative untreated controls calculating Abbott’s formula. Sicilian grape cultivars Nero d’Avola and Inzolia were evaluated and disease levels compared at the end of crop cycle to observe different responses to treatments and/or disease susceptibility. Since ranking of treatments efficacies/effects was similar for both trials conducted in the two vineyards, diseases, yield, and chemical and microbiological data were averaged for at least two trials and reported for each cultivar.

#### 2.1.1. Climate Data

Climatic conditions favorable to powdery mildew were detected in the 2020 season. In Figure 1, the mean weekly values of air temperature, relative humidity and rain are reported by averaging raw data from 1 April up to 30 September. Moreover, in this Figure, phenological stages of vine budbreak (from 25 to 31 March), bloom and veraison are indicated.

#### 2.1.2. Efficacy in Controlling Powdery Mildew Infections in Vineyards

Effects of two single factors, treatment and cultivar, were always significant for all the tested parameters, whereas site effects were not significant (Table 1). Therefore, the data clearly showed a different susceptibility to the powdery mildew of grape between the two tested cultivars, being Nero d’Avola more susceptible than Inzolia cultivar (Table 1, Figure 2). All first and second order interactions were not significant versus disease parameters except for treatment × cultivar on DS and I_MK_ variables (Table 1).

Based on the ANOVA results, the trials were analyzed separately for Nero d’Avola and Inzolia cultivars. Post-hoc analyses to establish the ranking of effectiveness at the end of crop cycle are reported in the Table 2.

These data showed that no significant DI differences were observed among different treatments on Nero d’Avola, whereas these differences were detected for Inzolia cultivar, and in this case the Cu–S complex and Cu–S complex plus mycorrhiza combination were the only effective treatments (significant data). Otherwise, significant differences were always observed among DS and I_MK_ values both on Nero d’Avola and Inzolia. In detail, the Cu–S complex and Cu–S complex plus mycorrhiza combination were the most effective treatments being able to significantly reduce powdery mildew DS and I_MK_ values by about 56–62% on Nero d’Avola and about 43–48% on Inzolia. Although to a lesser extent, DS and I_MK_ were significantly decreased by COS-OGA plus mycorrhiza treatment both on Nero d’Avola and Inzolia with percentage reductions of about 15% and 33%, respectively. COS-OGA applied alone significantly reduced about by 28% DS and I_MK_ values on Inzolia cultivar.

#### 2.1.3. Efficacy in Controlling Gray Mold Infections in Vineyards

The effects of single factors, treatment and cultivar were always significant on both DI, DS and I_MK_ parameters, whereas the effects of the site were always not significant. All interactions between factors were not significant on DI, DS and I_MK_ except for treatment × cultivar on DS parameter (Table 3). Therefore, Nero d’Avola revealed higher susceptibility degree to gray mold infection (Figure 3) than Inzolia cultivar (Table 3).

Based on ANOVA results, the trials were analyzed separately for Nero d’Avola and Inzolia cultivar. Post-hoc analyses showed the same ranking of treatment effectiveness for Nero d’Avola and Inzolia cultivars (Table 4).

On Nero d’Avola, significant differences were detected among treatments for DI parameter. In detail, Cu–S complex, Cu–S complex plus mycorrhiza, COS-OGA and COS-OGA plus mycorrhiza treatments significantly reduced the DI variable with values comprised between 50% and 75%. All treatments significantly reduced DS and I_MK_ values from 73% up to 85% if compared to the untreated control according to Abbott’s formula. Moreover, on Inzolia grape all treatments inhibited gray mold development.

#### 2.1.4. Efficacy in Controlling Sour Rot Infections in Vineyards

Similar to previous experiments, the effects of the single factors, treatment and cultivar, were significant on DI, DS and I_MK_ parameters, whereas the effects of the site were always not significant. All interactions between factors were not significant on all disease parameters except for treatment × cultivar on DS (Table 5). Nero d’Avola revealed higher susceptibility degree to sour rot infection (Figure 4) than Inzolia cultivar (Table 5).

Post-hoc analyses revealed a similar ranking of efficacy on two wine grape cultivars except for DS and IMK parameters on Nero d’Avola (Table 6). On this cultivar, all treatments were significantly effective to reduce the number of infected bunches compared to the untreated control. In detail, the most effective treatment was once again the Cu–S complex plus mycorrhiza combination, since it reduced DI, DS and IMK values by approximately 72–82% if compared to untreated controls (Abbott’s formula). Although with slightly lower performances, COS-OGA and COS-OGA plus mycorrhiza were able to significantly reduce sour rot decay. On Inzolia, all treatments were significantly effective in reducing or inhibiting sour rot infections if compared to the control.

### 2.2. Disease Incidence and Severity Progressions of Powdery Mildew

Powdery mildew disease (DI and DS) progressions over time for each cultivar are shown in Figure 5. On Nero d’Avola, the DI value was averagely very high and it reached maximum level in July (DI = 100%) in untreated control plots, whereas DS value reached the maximum in August (DS = 3.5). Likewise, powdery mildew infections detected on Inzolia grape showed averagely high DI values. In detail, high decay amounts were already observed starting from June and increased at the following evaluations. Unlike Nero d’Avola, on Inzolia the higher level of DI was reached in August (97.5%), whereas DS maximum levels were reached in September (DS = 2.15). Comprehensively, lower values of DS were recorded overt time on Inzolia vineyards if compared to Nero d’Avola, thus confirming the higher disease susceptibility of the red wine grape cultivar (Figure 5).

### 2.3. Yield and Chemical Analysis of Grape Must

The impact of sustainable compounds on grape yield and quality parameters of wine grape, including the total soluble solids (°Brix), total acidity (g L^–1^ of tartaric acid) and pH, was assessed (Table 7).

Treatments provided significant effects only on yields of Nero d’Avola and Inzolia wine grapes (Table 7). Nero d’Avola production always increased significantly in all treated plots if compared to the untreated control. Cu–S complex and Cu–S complex plus mycorrhiza combination provided the best performances with the highest production increases (about by 133%) if referred to relative control (Table 7). Similarly, the Cu–S complex and Cu–S complex plus mycorrhiza combination increased wine grape yield by 153% on Inzolia vineyard (Table 7). Although with lower performances, also the COS-OGA and COS-OGA plus mycorrhiza combination significantly increased the average yields of Nero d’Avola (from 50 to 67%) and Inzolia (from 92 to 113%) wine grapes if compared to those of untreated controls. The chemical analysis of the wine grapes showed that the sugar content, total acidity and pH were not significantly influenced by all treatments with respect to those recorded on Nero d’Avola and Inzolia untreated controls (Table 7).

### 2.4. Microbiological Analysis of Grapes

Four carposphere microbial communities were separately evaluated for each cultivar (fungal and yeast populations, and aerobic bacterial and fluorescent bacteria populations).

The cultivable fungal and yeast populations on Nero d’Avola and Inzolia berries are reported in Figure 6a and 6c, respectively. The fungal load detected on Nero d’Avola carposphere ranged from 3.44 to 3.61 Log_10_ CFU g^–1^, while the yeast load was comprised between 4.18 and 5.26 Log_10_ CFU g^–1^ throughout all treatments. The fungi load on Inzolia carposphere ranged from 3.29 to 3.44 Log_10_ CFU g^–1^, while the yeast load from 4.57 to 5.17 Log_10_ CFU g^–1^. Fungal and yeast loads were not significantly influenced by tested treatments both on Nero d’Avola and Inzolia wine grapes if compared to the relative untreated controls. The cultivable aerobic and fluorescent bacterial populations are reported in Figure 6b,d, respectively, for Nero d’Avola and Inzolia. The aerobic bacteria load on Nero d’Avola carposphere ranged from 4.75 to 5.35 Log_10_ CFU g^–1^, while the fluorescent bacteria from 2.41 to 3.73 Log_10_ CFU g^–1^ throughout all treatments. The aerobic bacteria load on Inzolia was comprised between 3.44 and 5.33 Log_10_ CFU g^–1^, while the fluorescent bacteria load ranged from 3.37 to 3.44 Log_10_ CFU g^–1^. The fluorescent bacteria load was not significantly influenced by the tested treatments both on Nero d’Avola and Inzolia grape. Otherwise, only for aerobic bacteria load significant differences were detected on different treated Inzolia wine grape (Figure 6d). In particular, in COS-OGA treatments (alone and in combination with mycorrhiza) aerobic bacteria load was similar to untreated control (not significant data); diversely Cu–S complex and Cu–S complex plus mycorrhiza combination significantly decreased the bacteria load.

## 3. Discussion

Organic viticulture is currently dependent on the availability of Cu and S, which are crucial components of grapevine protection against the main diseases. The massive use over time of Cu and S has led to negative effects on environment health. Therefore, in the present paper, sustainable and ecofriendly alternative compounds were tested for the first time in Sicily against powdery mildew, gray mold and sour rot under field conditions on Nero d’Avola and Inzolia, two among the most worldwide appreciated Sicilian cultivars. The effectiveness of COS-OGA as RI, and its effects combined with arbuscular mycorrhizal fungi (AMF) were assessed in controlling grapevine diseases. The good exposure of the Sicilian organic vineyards to solar radiation and wind reduced humidity conditions, thus limiting *P. viticola* infections. Otherwise, the climatic conditions were very conducive for powdery mildew epidemic, confirming that Sicily is a high-risk area for *E. necator* infections. Moreover, powdery mildew occurred with different disease pressures depending on the cultivar and, specifically, Nero d’Avola was most susceptible. On this cultivar whitish powdery efflorescence was associated with necrotic reticulation and suberization of the epidermal cells, often leading to berry cracking. Otherwise, a lower severity was recorded on the Inzolia cultivar. The symptoms consisted of necrotic reticulations, which generally do not evolve into berry cracks. Concerning gray mold and sour rot, the rainfalls that occurred from middle July up to early September (BBCH 81–89) resulted in low (Inzolia) and high (Nero d’Avola) disease pressures. As a consequence of berry cracking, Nero d’Avola was also severely affected by gray mold and sour rot. Diversely, lower levels of gray mold and sour rot decays have been recorded on Inzolia, reduced about to one third if compared to red wine grape cultivar. To the best of our knowledge, this study reports for the first time that Nero d’Avola is more susceptible to powdery mildew, gray mold and sour rot than Inzolia cultivar.

Although the Cu–S complex plus mycorrhiza and Cu–S complex always provided the best results against major fungal diseases of wine grapes, COS-OGA-based treatments gave also noteworthy performances. In detail, COS-OGA combined with mycorrhiza application followed by COS-OGA alone were effective in reducing severity of powdery mildew attacks on Inzolia, whereas only COS-OGA plus AMF combination were able to significantly reduce the powdery mildew amount on Nero d’Avola. Our data showed that the elicitor was more effective in reducing severity than incidence and AMF could have mitigated *E.*
*necator* infections. Since the colonization ability of AMF was not evaluated in this paper, further studies should be performed to confirm the performances of this treatment combination. However, present results are in accordance with findings obtained by van Aubel et al. [26] in French and Spain vineyards. This is probably due to the elicitors action mode, that do not involve direct toxic effects against pathogens but triggers natural host defenses, leading to a reduction of disease amounts; this could be the reason why disease incidence is less well-controlled than severity under high disease pressure [26]. In addition, COS-OGA applied alone and combined with mycorrhizal fungi always proved to be effective against gray mold and sour rot both on Nero d’Avola and Inzolia. In particular, on Inzolia their performances were comparable with Cu–S complex, which is the standard product for many organic wine growers.

Little is known about the effects of the organic vineyard management on the winemaking process [1], but frequent Cu and S treatments could probably compromise the composition and the sensory properties of the wine [9,35]. Moreover, Cu and S repeated applications in vineyards could influence the species diversity of fermentation microbiota [7,36], including indigenous yeasts and other environment-related microorganisms, which can contribute to define wine regional typicality [1]. Definitely, the replacement or reduction in chemical inputs by using of sustainable and eco-friendly compounds might lead to benefits for the postharvest stages and microbial communities. To this regard, as the present study demonstrates, COS-OGA is able to reduce yield losses and simultaneously maintains the main chemical grape characteristics—defining technological maturity—in full respect of carposphere microorganisms. Our findings showed that oenological parameters (sugar content, pH and total acidity) were not affected by the tested treatments, including COS-OGA application in agreement with previous data reported by Rantsiou et al. [7]. These authors evaluated the effect of chemical products, bioproducts and RIs (COS-OGA mixed with Cu and metiram) on yield and parameters involved in the technological and phenolic maturity of the grapes at the harvest time. Unlike Rantsiou et al. [7], the grape production varied considerably among the tested products being COS-OGA-based treatments always able to enhance the average yields and to preserve microbial communities on Nero d’Avola and Inzolia wine grapes. Moreover, COS-OGA-based applications did not negatively influence aerobic bacteria communities on Inzolia carposphere, whereas Cu–S-based treatments failed. Based on these data, COS-OGA application can be encouraged on a large scale since this compound does not exert negative pressure on carposphere microorganisms.

Although RIs are rarely used in viticulture, due to variable efficacy depending on several biotic and abiotic factors [23], COS-OGA compounds present many additional advantages if compared to Cu and S-based products. Their application, for example, does not imply any reentry interval for growers in vineyard, pre-harvest interval time, residues harmful for final consumers or the arising risk of resistance phenomena [26]. Moreover, the protection provided by elicitors as COS-OGA, is not specific and can potentially manage a wide spectrum of targeted phytopathogens as reported in the literature [37]. Although costs relative to COS-OGA applications are almost comparable to those reported for Cu–S, slight differences in the cost–benefit evaluation are justified by the above reported positive aspects. These data should be confirmed under different operative conditions. However, COS-OGA alone or combined with other organic control measures could be proposed to enhance the sustainability of the management of main airborne diseases and grape production. These efforts will allow a significant decrease over time in the use of Cu in organic viticulture according to the global green policies.

## 4. Materials and Methods

### 4.1. Field Experiments

Two field experiments were performed in duplicate during 2020 in two 10-year-old organic vineyards of *Vitis vinifera* L., i.e., on red berried Nero d’Avola and on white berried Inzolia wine grape cultivars, both grafted onto 140 Ru (Ruggeri) rootstock. Two trials for each cultivar were conducted in different sites of vineyards located at Rodì Milici (Messina province, Italy, lat. 38°06′ N; long. 15°08′ E, altitude of about 100 m a.s.l). The plants were spaced by 1 m in the rows, with 2.5 m between the rows, and they were grown according to the Guyot trellis system, leaving 5–6 buds per grapevine, with grass cover between the rows. The height of the fruiting cane was about 60–65 cm from the soil surface. The vineyards were not irrigated and the natural organo-mineral fertilizer (Vigna Pro, NPK 3-6-12, TerComposti S.p.A.) was distributed at the rate of 500 kg ha^–1^ banded under the grapevine in the winter, according to the common practices for the cultivation area. Moreover, arbuscular mycorrhizal fungi (Table 8) were applied as soil treatment at vine budbreak of the previous crop seasons (in 2018, 2019 and 2020).

A randomized complete block design with 4 replicates each consisting of 12 plants for treatment was adopted for each of the 4 experimental trials. Moreover, buffers were always inserted among plots differently treated. Four treatments for each cultivar were included and compared with an untreated control. All treatments were applied from the start of May until the end of August, with a total of 6 applications for each site (Table 8). At the time of the first application, grapevines were at the phenological stage of inflorescences swelling (BBCH 55) and the shoots were about 20 cm long. The COS-OGA and Cu–S applications were done by spraying a volume equivalent to 500 L ha^–1^, using a backpack sprayer (Volpi UNI mod. 78P) in treatments 1, 2, 3, 4 and by airblast sprayer in treatments 5 and 6, whereas AMF was sprayed only once onto the canopy (Table 8). Grapes were harvested at their optimum technological maturity (8 September). Thereafter, Nero d’Avola grapes were destemmed, left to macerate, and then crushed, whereas Inzolia ones were only destemmed and crushed.

### 4.2. Climate Data

The weather parameters, i.e., average temperature (°C), relative humidity (%) and rainfall (mm) were obtained from the data provided by the weather stations of Novara di Sicilia and Patti (Messina province) of Servizio Informativo Agrometeorologico Siciliano (SIAS), Sicily region. These data were implemented with phenological stages of vine budbreak, bloom and veraison.

### 4.3. Assessment of Disease Symptoms

Disease incidence and severity were determined by the assessment of symptoms in each plot of vineyards. Powdery mildew (*E. necator*), gray mold (*B. cinerea*) and sour rot (yeasts plus acetic acid-producing bacteria) [5,38] infections were directly evaluated on bunches. Powdery mildew infection was assessed at five different monitoring times (12 April, 15 June, 6 July, 5 August and 8 September, 2020), whereas gray mold and sour rot infections were assessed in September 2020 at the grape harvesting time. Disease symptoms were evaluated according to general EPPO guidelines [39] on four bunches for each plant (five plants for each plot). A five-point scale was used for both diseases with class ‘0’ being no symptoms and class ‘4’ being the highest damage. Class values corresponded to percentage infections range on the bunches—where class 0 = 0%, class 1 = 1–25% of infected berries on single bunch, class 2 = 25.1–50% of infected berries on single bunch, class 3 = 50.1–75% of infected berries on single bunch, class 4 = 75.1–100% of infected berries on single bunch. Data processing involved the calculation of the percentage of symptomatic bunches on the total number of examined bunches (= disease incidence, DI) and the average class (weighted mean) value of examined bunches (= disease severity, DS) for each plot. Moreover, the infection index (or McKinney’s index = I_MK_), which combines both the incidence and severity of the disease, was calculated according to the following equation:$$I_{MK}=\frac{\sum_{\;}\left(d\;\times \;f\right)}{N\times D}\;\times 100.\;$$
where *d* = category of disease class scored for the grape bunches; *f* = disease frequency; *N* = total number of examined bunches; *D* = highest class of disease intensity that occurred on the empirical scale [40].

Following disease assessment, a representative number of bunch samples were recovered randomly within each plot and transferred to the laboratory to identify and confirm the causal agents using microscope (Olympus–Bx61, Tokyo, Japan) observation (hyphae and conidia for powdery mildew and greyish layers containing conidiophores and conidia for gray mold) and isolation (producing typical colonies of *B. cinerea*) onto potato dextrose agar (PDA, Oxoid, Basingstoke, UK). The streaking technique was used to recover yeasts and acid acetic-producing bacteria (responsible of sour rot) from macerated berries on yeast peptone dextrose agar (YPDA, 10 g L^–1^ of yeast extract, 10 g L^–1^ of peptone, 20 g L^–1^ of dextrose, 20 g L^–1^ of bacteriological agar) supplemented with 100 mg L^–1^ of chloramphenicol and nutrient agar (NA, Oxoid, Basingstoke, UK) supplemented with 100 mg L^–1^ cycloheximide (sour rot).

### 4.4. Chemical Analysis of Grape Must

The quality parameters of must, including the total soluble solids (°Brix), total acidity (g L^–1^ of tartaric acid) and pH were determined through laboratory analysis carried out by Istituto Regionale del Vino e dell’Olio (IRVO) (Milazzo, Italy). Berry sampling was performed in a random way for all replicates in order to increase their representativeness, accordingly to the exposure and position of the berries on the bunch. The analyses were performed with the OenoFoss^TM^ instrument (FOSS Italia S.r.l. PD, Italy). Moreover, the average grape yield (weight of harvested bunches expressed as kg per plant) per treatment was recorded at harvest time.

### 4.5. Microbiological Analysis of Grapes

Carposphere microorganisms were evaluated on berries collected at harvest time. From each sample, approximately 100 g of healthy berries for each treatment were randomly removed from the bunches, placed in sterilized flasks with 500 mL of buffered peptone water (BPW, pH = 7.0 ± 0.2 at 25°C, Biolife, Milan, Italy) and 0.02% Tween 80 (VWR Chemicals, Solon, Ohio, USA), and then subjected to orbital shaking at 150 rpm for 1 h. Culturable bacteria, fungi and yeasts were assessed by plating tenfold serial dilutions in triplicate on four culture media: PDA and YPDA supplemented with 100 mg L^–1^ of chloramphenicol (AppliChem GmbH, Darmstadt, Germany) to inhibit bacterial growth; NA and King’B agar [41] supplemented with 100 mg L^–1^ cycloheximide (AppliChem GmbH, Darmstadt, Germany) to inhibit yeast and mold growth. Four different microbial communities were studied: total aerobic bacteria, fluorescent bacteria, fungi and yeasts. After incubation at 25°C for 2–5 days, colony forming units (CFUs) per unit of berry weight (CFUs g^–1^) were calculated.

### 4.6. Statistical Analyses

Data from field trials were subjected to analysis of variance by using the Statistical 10 package software (StatSoft Inc., Tulsa, OK, USA) to determine significant differences among the tested treatments in field performances against *E. necator*, *B. cinerea* and sour rot. Data obtained from two trials were compared and analyzed for each cultivar. Initial analyses of disease incidence (DI), severity (DS) and McKinney’s index (I_MK_) were conducted by calculating F and the correspondent P-value associated with the main source of variation (treatment, evaluation time, and cultivar) and with the interactions among them. Thus, arithmetic means of DI, DS and I_MK_ of the two trials for each cultivar were calculated, averaging the values determined for the single replicates of treatments. Percentage data (DI and I_MK_) were previously transformed using the arcsine transformation (sin^–1^ square root x). Post-hoc comparisons among different treatments were achieved by means of Fisher’s least significant difference test at α = 0.05. Similarly, significant differences in yield, chemical characteristics and microbial communities of wine grape were also assessed. The effects of the treatments on all tested parameters were also referred to the untreated control by using the Abbott’s formula [42].
$$I\left(\%\right)=\frac{C-T}{C}\;\times \;100$$
where *I* = percentage reduction data, *C* = mean parameter value in the untreated control plots and *T* = mean parameter value in the treated plots.

## 5. Conclusions

Comprehensively, COS-OGA applications could be considered as ecofriendly alternatives to Cu and S treatments since it manages natural infections of botrytis bunch and sour rots and, to a lesser extent, those of powdery mildew occurring in organic vineyards. For the latter disease, their performances depend on disease pressure and cultivar susceptibility. Although on average, less effective than Cu-S applications, COS-OGA alone or combined with AMF is able to reduce yield losses and to maintain postharvest grapes quality. Nevertheless, a further validation of this treatment combination is necessary since herein the colonization by AMF was not evaluated. Under high-disease pressure, COS-OGA represents an option to be integrated with already existing organic measures since it has a wide pathogen spectrum; no negative effects on carposphere microbiota; no harmful implications for environment, farmers or consumers; no residue and no fungicide resistance risks in total respect of current GND policies.

## Figures and Tables

**Figure 1 plants-11-01763-f001:**
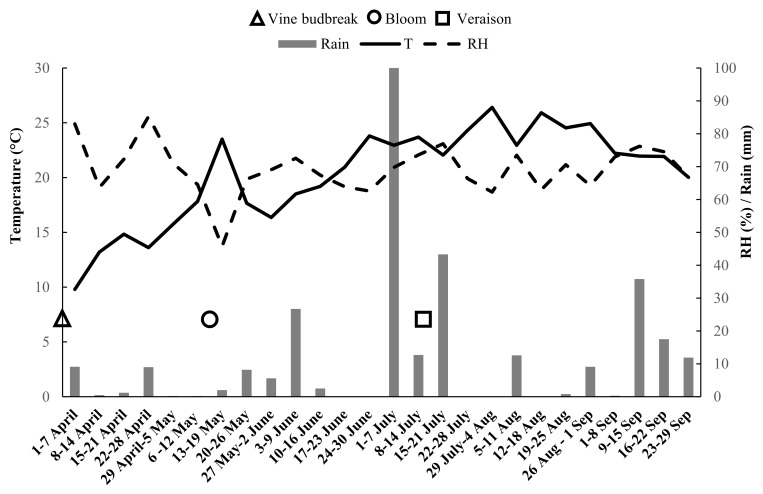
Climate data and main phenological stages of grapevines detected in the 2020 season, from April to September, by the weather stations of Novara di Sicilia and Patti (ME). T = temperature; RH = relative humidity.

**Figure 2 plants-11-01763-f002:**
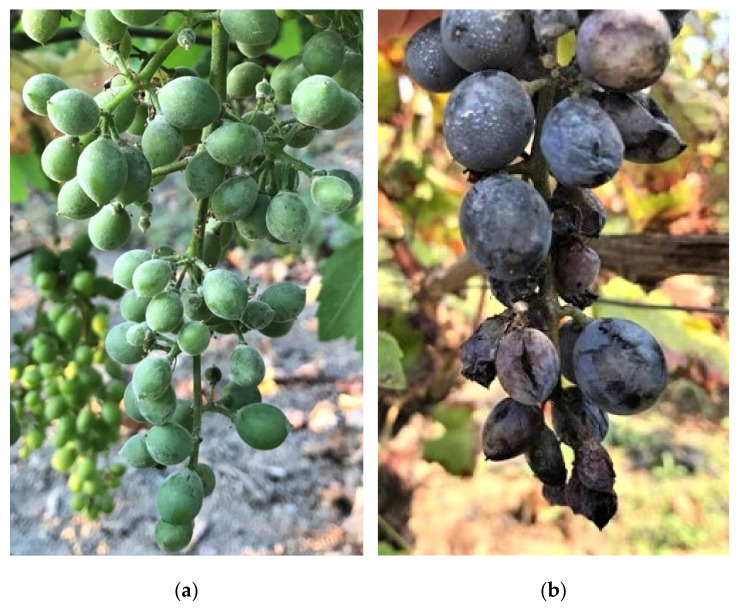
Symptoms observed on grapes caused by *Erysiphe necator.* Whitish powdery efflorescence (**a**) and berry cracks (**b**) on bunches of Nero d’Avola.

**Figure 3 plants-11-01763-f003:**
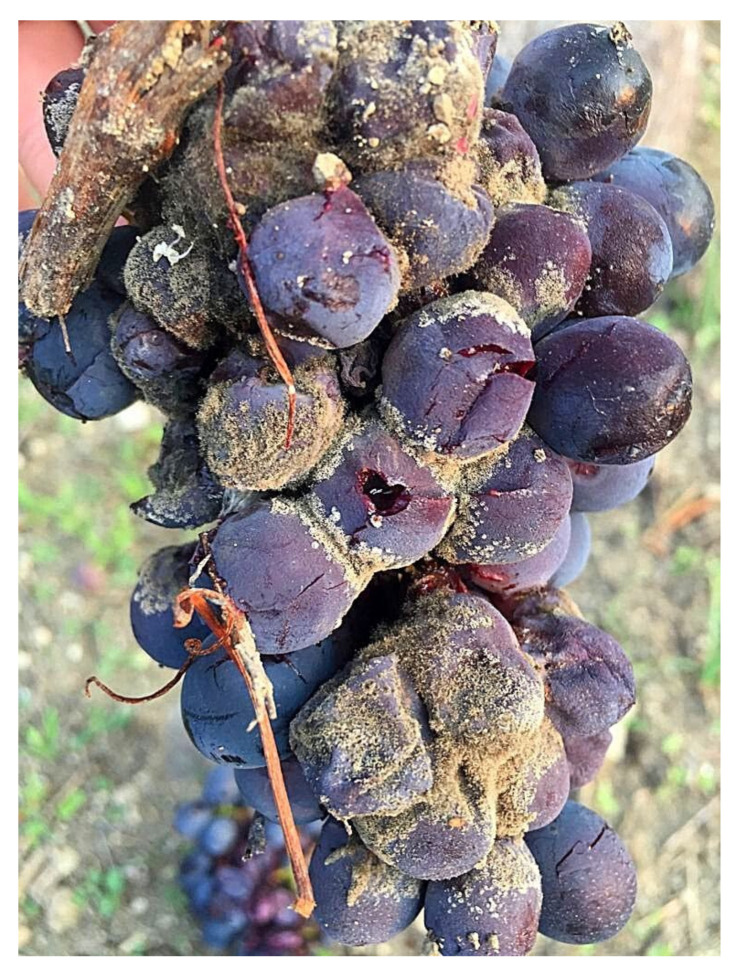
Symptoms observed on bunches of Nero d’Avola caused by *Botrytis cinerea*.

**Figure 4 plants-11-01763-f004:**
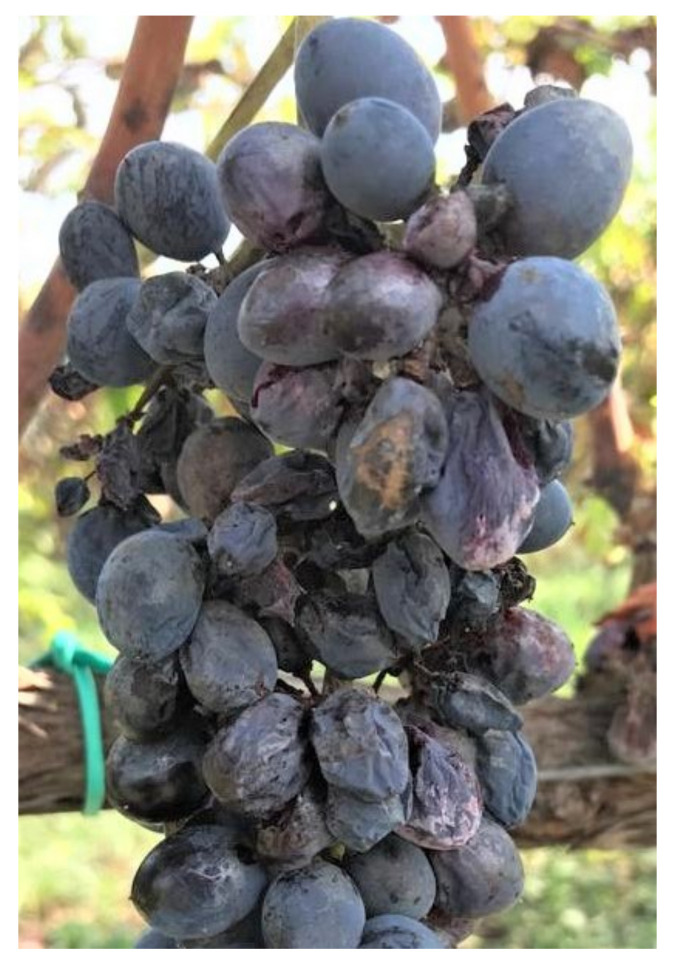
Symptoms observed on bunches of Nero d’Avola caused by sour rot.

**Figure 5 plants-11-01763-f005:**
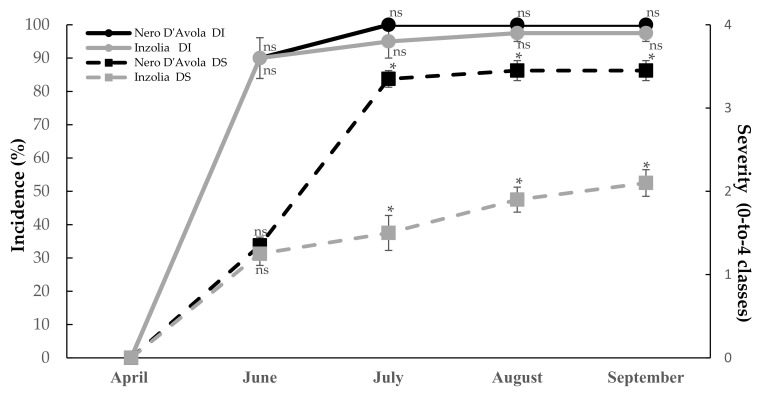
Progression over time of powdery mildew infections caused by *Erysiphe necator* in control plots of vineyards. Continuous lines indicate dual comparisons of DI progression values (percent) over time whereas dotted lines show dual comparisons of DS progression values (0-to-4 scale) over time between Nero d’Avola (black lines) and Inzolia (grey lines). ns = not significant; * = significant differences.

**Figure 6 plants-11-01763-f006:**
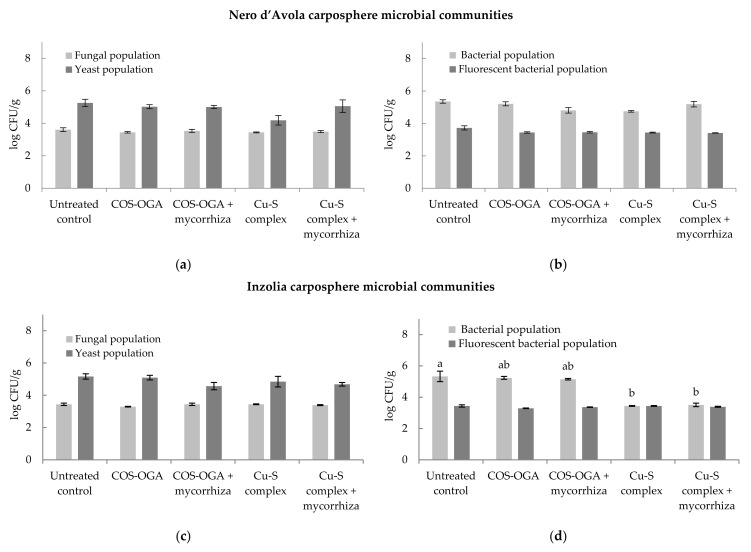
Fungal and yeast population on (**a**) Nero d’Avola and (**c**) Inzolia carposphere. Bacterial and fluorescent bacterial population on (**b**) Nero d’Avola and (**d**) Inzolia carposphere. Data presented as Log_10_ CFU/g of fresh weight. Bars show the standard error of the mean (±SEM). Bacterial population data followed by different letter(s) differs significantly according to Fisher’s least significance differences test (α = 0.05).

**Table 1 plants-11-01763-t001:** Effects of single factors and their interactions in ANOVA on the powdery mildew infection on wine grape caused by *Erysiphe necator* over time.

		Disease Incidence (DI)		Disease Severity (DS)		McKinney’s Index (I_MK_)	
Source of Variation	df	F	*p*-Value	F	*p*-Value	F	*p*-Value
Treatment	4	4.909	*0.001366*	72.439	*<0.0001*	59.012	*<0.0001*
Cultivar	1	11.879	*0.000908*	206.501	*<0.0001*	173.176	*<0.0001*
Site	1	2.182	0.143575 ^ns^	1.084	0.300866 ^ns^	1.384	0.242915 ^ns^
Treatment × Cultivar	4	0.364	0.833815 ^ns^	17.995	*<0.0001*	15.512	*<0.0001*
Treatment × Site	4	0.061	0.993079 ^ns^	0.024	0.998849 ^ns^	0.093	0.984535 ^ns^
Treatment × Cultivar × Site	4	0.242	0.913439 ^ns^	0.029	0.998354 ^ns^	0.042	0.996549 ^ns^

*p*-value of fixed effects associated to F test; *ns*: not significant data.

**Table 2 plants-11-01763-t002:** Post-hoc analyses of treatment effects on disease incidence (DI), severity (DS) and McKinney’s index (I_MK_) of powdery mildew on Nero d’Avola and Inzolia wine grapes caused by *Erysiphe necator* at the final production stages (on September 8th).

Treatment	Nero d’Avola ^a,b^			Inzolia ^a,b^		
	DI (%)	DS (0-to-4)	I_MK_ (%)	DI (%)	DS (0-to-4)	I_MK_ (%)
Untreated control	100 ± 0.0 ^ns^	3.4 ± 0.11 a	86.2 ± 2.90 a	97.5 ± 2.5 a	2.1 ± 0.15 a	52.5 ± 3.88 a
Cu–S complex	92.5 ± 5.0	1.3 ± 0.11 c	33.1 ± 2.72 c	82.5 ± 5.0 b	1.1 ± 0.11 c	26.9 ± 2.72 c
Cu–S complex + mycorrhiza	92.5 ± 5.0	1.5 ± 0.18 c	37.5 ± 4.41 c	82.5 ± 5.0 b	1.2 ± 0.10 bc	28.8 ± 2.50 bc
COS-OGA	100 ± 0.0	3.2 ± 0.23 ab	81.2 ± 5.85 ab	92.5 ± 5.0 ab	1.5 ± 0.21 b	38.8 ± 5.28 b
COS-OGA + mycorrhiza	100 ± 0.0	2.9 ± 0.18 b	73.1 ± 4.59 b	92.5 ± 5.0 ab	1.4 ± 0.12 bc	35.0 ± 3.03 bc

^a^ Data expressed as means of the two trials and followed by standard error of the means (± SEM). Each value derives from 5 replicates, each formed by at least 4 bunches. ^b^ Arcsine transformation was used on percentage data prior to analysis, whereas untransformed data (%) are presented. DI, DS and I_MK_ values followed by the same letter within each column are not significantly different according to Fisher’s least significance differences test (α = 0.05). DI = Disease incidence; DS = disease severity; I_MK_ = McKinney’s index; ns = not significant data.

**Table 3 plants-11-01763-t003:** Effects of single factors and their interactions in ANOVA on gray mold caused by *Botrytis cinerea* on wine grape.

		Disease Incidence (DI)		Disease Severity (DS)		McKinney’s Index (I_MK_)	
Source of Variation	df	F	*p*-Value	F	*p*-Value	F	*p*-Value
Treatment	4	17.8744	*<0.0001*	35.4128	*<0.0001*	27.6946	*<0.0001*
Cultivar	1	778291	*<0.0001*	95.1193	*<0.0001*	96.7338	*<0.0001*
Site	1	0.1608	0.689486 ^ns^	1.1743	0.281772 ^ns^	0.6645	0.417390 ^ns^
Treatment × Cultivar	4	0.8693	0.486126 ^ns^	12.4587	*<0.0001*	1.6885	0.160808 ^ns^
Treatment × Site	4	0.0352	0.997585 ^ns^	0.0734	0.990027 ^ns^	0.0633	0.992476 ^ns^
Treatment × Cultivar × Site	4	0.0754	0.989510 ^ns^	0.0183	0.999327 ^ns^	0.0609	0.993018 ^ns^

*p*-value of fixed effects associated to F test; *ns*: not significant data.

**Table 4 plants-11-01763-t004:** Post-hoc analyses of treatment effects on disease incidence (DI), severity (DS) and McKinney’s index (I_MK_) of gray mold caused by *Botrytis cinerea* on Nero d’Avola and Inzolia wine grapes.

	Nero d’Avola ^a,b^			Inzolia ^a,b^		
Treatment	DI (%)	DS (0-to-4)	I_MK_ (%)	DI (%)	DS (0-to-4)	I_MK_ (%)
Untreated control	70 ± 9.35 a	1.1 ± 0.16 a	27.5 ± 4.12 a	22.5 ± 2.5 a	0.2 ± 0.02 a	5.6 ± 0.62 a
Cu–S complex	17.5 ± 7.51 b	0.2 ± 0.07 b	4.4 ± 1.87 b	0.0 ± 0.0 b	0.0 ± 0.0 b	0.0 ± 0.0 b
Cu–S complex + mycorrhiza	22.5 ± 10.0 b	0.2 ± 0.10 b	5.6 ± 2.50 b	0.0 ± 0.0 b	0.0 ± 0.0 b	0.0 ± 0.0 b
COS-OGA	35 ± 10.0 b	0.3 ± 0.08 b	7.5 ± 2.12 b	0.0 ± 0.0 b	0.0 ± 0.0 b	0.0 ± 0.0 b
COS-OGA + mycorrhiza	25 ± 13.69 b	0.2 ± 0.11 b	5.6 ± 2.86 b	0.0 ± 0.0 b	0.0 ± 0.0 b	0.0 ± 0.0 b

^a^ Data expressed as means of the two trials and followed by standard error of the means (± SEM). Each value derives from 5 replicates, each formed by at least 4 bunches. ^b^ Arcsine transformation was used on percentage data prior to analysis, whereas untransformed data (%) are presented. DI, DS and I_MK_ values followed by the same letter within each column are not significantly different according to Fisher’s least significance differences test (α = 0.05). DI = Disease incidence; DS = disease severity; I_MK_ = McKinney’s index; ns = not significant data.

**Table 5 plants-11-01763-t005:** Effects of single factors and their interactions in ANOVA on the sour rot infection caused by phytopathogenic bacteria and yeasts on wine grape.

		Disease Incidence		Disease Severity		McKinney’s Index (I_MK_)	
Source of Variation	df	F	*p*-Value	F	*p*-Value	F	*p*-Value
Treatment	4	44.8487	*<0.0001*	34.2016	*<0.0001*	34.1885	*<0.0001*
Cultivar	1	205.5921	*<0.0001*	154.9407	*<0.0001*	192.4708	*<0.0001*
Site	1	1.5921	0.210692 ^ns^	0.7905	0.376611 ^ns^	0.6381 ^ns^	0.426747 ^ns^
Treatment × Cultivar	4	2.2039	0.075924 ^ns^	9.3083	<0.0001	1.9823 ^ns^	0.105118 ^ns^
Treatment × Site	4	0.0461	0.995922 ^ns^	0.0198	0.999221 ^ns^	0.0276 ^ns^	0.998498 ^ns^
Treatment × Cultivar × Site	4	0.0855	0.986682 ^ns^	0.0277	0.998490 ^ns^	0.0599 ^ns^	0.993223 ^ns^

*p*-value of fixed effects associated to F test; *ns*: not significant data.

**Table 6 plants-11-01763-t006:** Post-hoc analyses of treatment effects on disease incidence (DI), severity (DS) and McKinney’s index (I_MK_) of sour rot caused by phytopathogenic bacteria and yeasts on Nero d’Avola and Inzolia wine grapes.

	Nero d’Avola ^a,b^			Inzolia ^a,b^		
Treatment	DI (%)	DS (0-to-4)	I_MK_ (%)	DI (%)	DS (0-to-4)	I_MK_ (%)
Untreated control	97.5 ± 2.5 a	1.7 ± 0.10 a	43.1 ± 2.50 a	37.5 ± 5.6 a	0.4 ± 0.07 a	10.6 ± 1.88 a
Cu–S complex	40.0 ± 10.7 b	0.5 ± 0.12 bc	11.2 ± 2.89 bc	0.0 ± 0.0 b	0.0 ± 0.0 b	0.0 ± 0.0 b
Cu–S complex + mycorrhiza	27.5 ± 11.4 b	0.3 ± 0.13 c	8.1 ± 3.37 c	0.0 ± 0.0 b	0.0 ± 0.0 b	0.0 ± 0.0 b
COS-OGA	47.5 ± 6.1 b	0.9 ± 0.25 b	22.5 ± 6.20 b	7.5 ± 7.5 b	0.07 ± 0.07 b	1.9 ± 1.87 b
COS-OGA + mycorrhiza	47.5 ± 6.1 b	0.6 ± 0.19 bc	15.6 ± 4.84 bc	2.5 ± 2.5 b	0.02 ± 0.02 b	0.6 ± 0.62 b

^a^ Data expressed as means of the two trials and followed by standard error of the means (± SEM). Each value derives from 5 replicates, each formed by at least 4 bunches. ^b^ Arcsine transformation was used on percentage data prior to analysis, whereas untransformed data (%) are presented. DI, DS and I_MK_ values followed by the same letter within each column are not significantly different according to Fisher’s least significance differences test (α = 0.05). DI = Disease incidence; DS = disease severity; I_MK_ = McKinney’s index; ns = not significant data.

**Table 7 plants-11-01763-t007:** Grape production and oenological parameters of Nero d’Avola and Inzolia.

	Nero d’Avola ^a,b^			
Treatment	Yield (kg plant^–1^)	Sugar Content (°Brix)	Total Acidity (g L^–1^)	pH
Untreated control	0.6 ± 0.04 d	20.8 ± 0.19 ^ns^	6.2 ± 0.08 ^ns^	3.1 ± 0.01 ^ns^
Cu–S complex	1.4 ± 0.03 a	21.0 ± 0.16	6.2 ± 0.03	3.2 ± 0.01
Cu–S complex + mycorrhiza	1.4 ± 0.03 a	20.6 ± 0.17	6.1 ± 0.07	3.3 ± 0.15
COS-OGA	0.9 ± 0.04 c	20.9 ± 0.22	6.0 ± 0.07	3.2 ± 0.01
COS-OGA + mycorrhiza	1.0 ± 0.05 b	20.8 ± 0.25	6.2 ± 0.05	3.3 ± 0.14
	Inzolia ^a,b^			
Untreated control	0.71 ± 0.02 d	19.88 ± 0.09 ^ns^	5.12 ± 0.11 ^ns^	3.36 ± 0.01 ^ns^
Cu–S complex	1.80 ± 0.08 a	19.52 ± 0.09	5.36 ± 0.15	3.46 ± 0.10
Cu–S complex + mycorrhiza	1.80 ± 0.06 a	19.82 ± 0.10	5.18 ± 0.08	3.42 ± 0.05
COS-OGA	1.36 ± 0.03 c	20.04 ± 0.08	5.00 ± 0.03	3.45 ± 0.05
COS-OGA + mycorrhiza	1.51 ± 0.04 b	19.93 ± 0.09	5.30 ± 0.04	3.43 ± 0.05

^a^ Data expressed as means of the two trials and followed by standard error of the means (±SEM). Each value derives from 5 replicates, each formed by at least 4 bunches. ^b^ Values followed by the same letter within each column are not significantly different according to Fisher’s least significance differences test (α = 0.05). ns = not significant data.

**Table 8 plants-11-01763-t008:** Integrated strategies adopted in the application of Cu and S formulation and alternative products (COS-OGA and mycorrhizae) and dates of the grapevine foliar applications for Nero d’Avola and Inzolia cultivars in different sites.

Treatment/Active Compound	Dosage	Product and Company	N. and Timing of Applications *
Untreated control	-	-	2 on May 2nd and 25th; 2 on June 15th and 30th; 1 on July 22nd and 1 on August 13th
COS–OGA	3 L ha^–1^	Ibisco^®^, Gowan Italia S.r.l.	
COS-OGA + mycorrhiza	3 L ha^–1^ + 5 kg ha^–1^	Ibisco^®^; Micosat F^®^ MO, CCS Aosta S.r.l.	
Cu–S complex 3%	5 L ha^–1^	Heliocuivre^®^–Heliosoufre^®^, CBC Europe S.r.l.	
Cu–S complex 3% + mycorrhiza	5 L ha^–1^ + 5 kg ha^–1^	Heliocuivre^®^–Heliosoufre^®^; Micosat F^®^ MO	

* Application data referred only to COS-OGA and Cu–S complex, while mycorrhiza (AMF) were applied only once on 5 June 2020.

## Data Availability

The data presented in this study are available on request from the corresponding author.

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
