# Peer review of "COS-OGA Applications in Organic Vineyard Manage Major Airborne Diseases and Maintain Postharvest Quality of Wine Grapes"

_plants, 2022, doi:10.3390/plants11131763_

Round 1
Reviewer 1 Report
Climate change and food security issues nowadays call for an eco-friendly method to control the pathogens responsible for crop loss. The authors evaluated the effectiveness of COS-OGA on treatment against powdery mildew, gray mold and sour rot under field conditions on Nero d’Avola and Inzolia Sicilian cultivars and reported a significant reduction of powdery mildew severity on both cultivar. Better results obtained when used in combination with AMF against gray mold and sour rot. This information is useful to grapevine farmers to better inform in the control measures of these pathogens in an organic agroecosystems thus fighting against environmental pollution from the traditional Cu and S compounds frequently used.
In view of ameliorating the scientific soundness for publication, I proposed some minor changes and suggestions.
1- Line 25: remove "on" in front of quality
2- Line 48: remove "the" before agro-ecosystem
3- Line 103: insert space between "2010" and (32)
4- Line 293: Species name in italics
5- The authors applied maceration in their treatments, however, this was limited only to one of the cultivar (Nero d'Avola) used. I find this being bias. Can this be clearly explain why this difference in treatment and support how it doesn't influence the outcome of disease or measurement of related parameters.
6- Table 9 can be modify to have four columns: Cultivar, Treatment, Formulation and Time.
7- Symptoms was assessed, it is of importance for the authors to include pictures of field symptoms and or cultured pathogens.
8- Why not include the contributors taxonomy information?
Author Response
Response to Reviewer 1 Comments
Point 1: Climate change and food security issues nowadays call for an eco-friendly method to control the pathogens responsible for crop loss. The authors evaluated the effectiveness of COS-OGA on treatment against powdery mildew, gray mold and sour rot under field conditions on Nero d’Avola and Inzolia Sicilian cultivars and reported a significant reduction of powdery mildew severity on both cultivar. Better results obtained when used in combination with AMF against gray mold and sour rot. This information is useful to grapevine farmers to better inform in the control measures of these pathogens in an organic agroecosystems thus fighting against environmental pollution from the traditional Cu and S compounds frequently used..
Response 1: Ok, thank you for the overall positive comment.
Point 2: In view of ameliorating the scientific soundness for publication, I proposed some minor changes and suggestions.
Response 2: Ok, we addressed point-by-point your ameliorating comments
1- Line 25: remove "on" in front of quality
Ok, done;
2- Line 48: remove "the" before agro-ecosystem;
Ok, done.
3- Line 103: insert space between "2010" and (32)
Ok, I add space definitely deleting the entire full reference “Cabrera et a. 2010” and maintained only number. Moreover, we replaced round brackets with square brackets for number reference herein and for the remaining references through entire manuscript as required by journal.
4- Line 293: Species name in italics;
Ok, done;
5- The authors applied maceration in their treatments, however, this was limited only to one of the cultivar (Nero d'Avola) used. I find this being bias. Can this be clearly explain why this difference in treatment and support how it doesn't influence the outcome of disease or measurement of related parameters.
Thank you for the comment. A different vinification protocol between Nero d’avola (redberries grape) and Inzolia (whiteberries grape) according to the common and traditional winemaking procedures adopted in Italy (red and white vinification, respectively). Being postharvest processing, authors believe these procedures did not influence disease and other parameters.
6- Table 9 can be modify to have four columns: Cultivar, Treatment, Formulation and Time.
Ok, Thank you for the precious suggestion. Done (see revised MS).
7- Symptoms was assessed, it is of importance for the authors to include pictures of field symptoms and or cultured pathogens.
Ok, Thank you for the suggestion. Three explicative photos were added in the revised MS.
8- Why not include the contributors taxonomy information?
Thank you for the comment. Since the causal agents are well known and easy recognizable based on symptomatology, the authors think taxonomy information is not necessary. However, preliminary morphological identification under microscope and numerous fungal and yeast isolation attempts were performed as reported in the M&M section (see revised manuscript).

Reviewer 2 Report
For specific comments please refer to the attached file, I stopped grammar corrections and terminology. These need to be proofed.
The work conducted is very thorough, albeit just for one year. Further validation will be required. Especially as there was low disease risk for botrytis and sour rot and high for mildew. Will products work if mildew risk is lower and botrytis higher?
I also feel the presentation of results in many repetitive tables can be condensed by omitting the ANOVA tables and mentioning the significant trends and interaction in the text (which is already largely done), and by combining individual disease tables into one for easier oversight.
I strongly support the need for alternatives in organic and conventional horticulture. I am aware that disease thresholds are important factors to consider in product efficacy and i felt the authors could discuss this more comparing their results with previous work. The cost of replacing sulphur and/or adding additional treatments should also be mentioned. Would the growers be happy to step away from sulphur? with the data shown i would not (yet) because of mildew
The AMF application onto canopy (?) is unclear. Did the AMF change?

Author Response
Response to Reviewer 2 Comments
Minor Comments
Point 1: For specific comments please refer to the attached file, I stopped grammar corrections and terminology. These need to be proofed.
Response 1: Ok, authors proofread entire manuscript and correctly addressed comments and suggestions
A few grammar/text corrections (as in bold) and/or comments
Line 71 – under ongoing testing and validation …
Ok, Done
Line 77 – so which one of the BCAs is the best? And for what pathogens? and how many applications are needed? If you review, do a ‘proper job’ and provide more detail please
Ok, rephrased.
Line 78 – In regards to biostimulants
Ok, done
Line 80 – In detail, AMF..: you list all the other used alternatives, but then jump without explanation onto AMF. It needs a connection between the previous and this sentence. This stipulates that AMF are the best or the only truly viable alternative? Is this what you claim indeed? You need to paraphrase and provide more published evidence then.
Ok, explained, rephrased and supported accordingly.
Line 81 – more sustainable than what??
Ok, I delete “more”
Line 86 – (23). Not (23)).
Thank you for the comment. Done. We also replaced round brackets with square brackets for number reference herein and for the remaining references through entire manuscript as required by journal guidelines.
Line 88 – so how effective was laminarin and Sacc. extracts and chitosan? Qualify and quantify your review statements please.
Thank you for the comment. Done. We addressed your comments (see revised text).
Line 90 – what constitutes ‘ good’ protection?
Ok, I rephrased this term (see revised MS).
Line 92 – the COS-OGA explantions need to come earlier, immediately after the abbreviation is used and not two sentences later, please restructure.
Thank you for the comment. We correctly addressed your comment.
Line 102 – remove author citation
Ok, done
Line 106 – little information, not few , the whole sentence needs correcting. As 7 and 26 worked on powdery in grape with the COS-OGA complex. And you want to build on that work and also the AMF work. So tell me more about the work by 7 and 26. And still waiting for the rationale why AMF and not PGRP or any of the others IRs, BCAs mentioned. You also chose three diseases, that probably could do with an introduction and how they affect organic wine production.
Ok, thank you for the comment i.e. little replacing few. We have chosen AMF accordingly to agronomic practice performed in commercial farm where the experiments were carried out. The authors focused also their attention on the three pathogens because they represent the key issues for the Mediterranean area (south Italy). To this regard, it is notherworthy underline the experiments were performed under natural infection pressure.
Line 361- italics for Vitis vinifera – really italics throughout and full name of pathogen for when first mentioned.but then I am reading M&M first, I see this was done
Ok, done throughout whole MS.
Line 366 – with grass cover between the rows (delete crop)
Ok, deleted (see rev. MS).
Line 367 – from the soil surface
Ok, done
line 368 – which fertilisers, what rate, and I know nothing about your common practices
Thank you for the comment. Done.
line 370 - four replicates consisting of 12 plants… any buffers between plants? what did the block consist of? 4 rows (one per replicate?)
Thank you, modification done. You are correct..blocks consist of 4 rows. Along single block the treated 12 plants were separated from the subsequent 12 plants differedly treated by an untreated plant and so on. In addition, between rows (blocks) an untreated row was also a buffer row.
line 371 – for each of the four
Ok, done
line 373 – what is a preventative criterion? I also assume these were applied not performed
Ok, modified and the suggestions addressed (see rev. MS).
line 377 – you need to add treatment number to the table if you refer to treatment numbers in the text. Basically I only see 4 treatments, so you mean application times not treatment!?
Ok, Thank you for your suggestion (see rev. MS). The table was improved to made it more readable and complete.
Table 9 – just list the applications dates and add year, I can count if there were two in a month. What is missing in the text or table the suppliers of the products used. Also please add to the table caption that all products, except the untreated control where applied on the same day (at least this is how I understand it)
Ok, Thank you for the comment. A column including product and company was added in the new ri-edited table.
Line 392 – cause of sour rot?
Ok, causal agents were added (see rev. MS).
Line 395 – at five different monitoring times
Ok, modification done (see rev. MS).
Line 395 – always add the year – probably something to be done throughout
Ok, modification done.
Line 412 – what is a representative sample, please specify
Ok, the sampling for chemical analysis of grape interested all replicates (and not only replicate) in vineyards of each treatment. Therefore, detected and reported value (e.g sugar content, pH and so on) is averagely representative of each treatment.
Line 413 – to the laboratory
Ok, modification done (see revised text).
414 – causal agent using the microscope. Then about here I would expect what structures you examined for mildew, botrytis and sour rot, followed by a separate sentence that in the absence of conidia botrytis mycelial pieces were isolated onto PDA (at least this is what I assume you have done). and what for sour rot? – some images of the disease as observed under the microscope or in the field would not go amiss with your field scoring
Ok, modifications were performed and some explaining sentences about sour rot added in the revised MS. Moreover, as you required we added some “in the field” images of sour rot as well as of powdery mildew and gray mold of grape.
420- how many berries? Or do you mean bunches? did you sample individual berries from a bunch in the field? How? And what do you mean by every 2-3 rows? you had only four reps and I don’t know what the block lay out and design was
Ok, samplig always referred to grape berries collecting from bunches belonging to all replicates (four reps) to reach about 200 g amount. This is the needed amount to determine oenological parameters for OenoFoss instrument. Please, about block design see considerations and clarifications previously reported.
424 – no idea what you mean by the grape production per treatment was recorded at harvest time? do you mean yield? Expressed as in number of bunches, berries per bunches and bunch or berry weight? Or total yield? Also describe the how please
Ok, thank for the clarification. We determine grape yield at harvest time and calculated as the average weight of harvested bunches per plant expressed as kg per plant. Data are averaged and obtained from all replicates (four reps).
427- 100g is how many berries? i doubt you will have gotten exactly 100g! And when was harvest time? I might have overlooked this? and how were bunches harvested and stored? Berries were snipped from the bunches ie with pedicel on or pulled from the rachis?
The number of berries to reach 100 g depends on cultivar. However, you are correct; 100 g is an approximate and indicative amount (see revised and modified MS).
439 – which agar was used for which enumeration? 100 g of smaller berries have a larger surface area than 100 g of large berries- have you looked at standardisation by berry umber and/or berry surface area?
Ok. The agar used was Oxoid. Although you are correct, we have not planned a standardization procedure since the comparisons among different treatments were always performed within the same cultivar (Nero d’Avola or Inzolia). However, before analysis we have chosen groups as homogenously as possible.
117 – I hate data presented solely relative to the untreated control. You MUST supply the actual disease levels for the controls too, to put it into context of the disease risk and pressure. The lack of being able to convert into ‘actuals’ to me is a complete reject of a manuscript. But looking at the tables this is actual DI and DS data, so what do you mean by relative? By relative I mean one standardises the control to 100%
Please, I want to point out that data of disease parameters (DI, DS and IMK) reported in Tables 2, 4 and 6 are raw data (i.e. actual and real disease data). However, the results referred to the above tables are discussed as reductions relative to untreated controls (i.e. according to Abbott’s formula). Is it perhaps not a widely reported notation for such as scientific papers focused on efficacy of control means?
Fig 1- I assume you have a Gubler model equivalent that tells you about disease risk, please add disease risk accordingly for ease of interpretation of the climate data. similarly I assume you have a botrytis risk model at hand? Is there one for sour rots?
Please, I want to point out that climatic data included in this paper are not related to some evaluations of disease risk occurrence for developing forecasting models. The paper is simply focused on activity of COS-OGA against powdery mildew, gray mold and sour rot of two wine grape cvs and the climatic can only support and complete our phytopathological findings.
Table 1 – I am also sure you have published or at least anecdotal evidence that these two cv vary in disease susceptibility to the three diseases monitored. You actually should add something into the intro or M&M to that effect
Please, my coauthors and I did not have clear evidence about different cultivar susceptibility to three key pathogens or have even published data about that up until now as you simply suppose. In this paper we do not rely on hypotetical assumptions but only to the rigorous methodologies and approaches including statistical analysis. In our opinion this is a new finding not reported until today and very useful and for winegrowers.
Table 2 – not surprised that DI is very similar and treatment differences are based on DS. That is to be expected. Having the actual disease levels will be important to understand if the data while numerically different is biological meaningful. And what the implication is for the growers between the different treatments outcomes. We will see what is in the discussion
I want to point out that DI is only a qualitative disease parameter (presence/absence of disease) whereas an accurate assessment must also rely on quantitative parameters (DS and Imk). Therefore, like you, I am not even surprised since that significant differences were detected for quantitative disease parameters. We think that data are adequately debated and addressed in the discussion focusing also on the implications for growers.
Anova tables – I don’t know if we really need them? I would think you can state this in the text, particulalry that overall you have no interactions and for each disease treatment and cultivar effect.
My coauthors and I believe that ANOVA tables are just as important as Post-hoc ones. Indeed, they underline not only the significance for treatment but also for cultivar factor and state not significativity for site and all referred interactions.
Results tables- I would like to see these as one table, mildew, botrytis and sour rot, stacked by disease, then I can see them at once without having to flick pages.
Thank you for the suggestion but my coauthors and I believe that tables cannot be condensed in one table since three target disease, two cultivars and three disease parameters for each of these were involved. We think the resulting table should be more confuse for the Plants reader.
Fig 2- I would have thought it is the area under the curve that gets compared for disease progression over time? why did you plot DI and DS instead of the index? Just wondering, I do prefer this over the index as an index ‘muddles the waters’ sometimes. I assume sour rot was only done/visible at harvest, but what about botrytis?
Thank you for the comment but the progression over time was considered only for powdery mildew since its typical epidemiological features allow to do this type of evaluation. In brief, this figure exclusively aims to compare disease pressure levels (DI and DS) between Nero’d’Avola and Inzolia at different monitoring times. Otherwise, for botrytis and sour rot occurring next at harvesting time in our grape vineyards it was not possible to report similar disease progression curves
Table 7 and 8 can be combined
Thank you for the comment. The tables were combined in one for a better readability (see revised MS)
Fig 3 – are the fluorescent bacteria part of the b graph?
Yes, they are.
306 – that answers my earlier question, which is surprising that this was not known not even based on industry experience.
I can share your doubt but my coauthors and I have not found previous clear evidence for this statement in literature. Anyway, we believe the statement reported here in this paper is strongly supported by scientific data for each of these diseases over exeperiments.
460 – as COS-OGA is really not doing too much to mildew, growers still will have to use sulphur? Is there much point to do sulphur, AMF and COS_OAG? And what is the cost–benefit? I think that is missing in your conclusion
We agree with you. To confirm this, we underlined in several parts of discussion, results and conclusion sections the best perfomances of Cu-S based applications if compared to COS-OGA ones especially for powdery mildew. We cannot know if the growers will continue to use the sulphur but surely they will have a further and valid alternative to be integrated with already exisisting biological measures. In regards to successive your comment, the cost-benefit evaluation for COS-OGA is comparable since the costs are similar but, unlike to Cu-S componds, elicitor do not interfere with carposhere microbial communities.
AMF question. I am not clear how the AMF were applied onto the canopy or ground? And there is no evidence if the AMF status of roots has changed because of this application… We would inoculate rootstocks with AMF to ensure colonisation and then there is still no guarantee that initial colonisation is enduring over space and time. This is missing in the discussion.
Thank you for the comment. You are correct, this information is missing (see revised MS). AMF were at first applied at ground level of all vineyards in the previous crop season (at vine budbreaking). However, the COS-OGA treatments reported in the table 8 (M&M section) were sprayed onto the canopy as recommended by label indications and only once in the plots where it was planned your use.
Also please this, while comprehensive, is just one year study, I feel you ned to add to your discussion that results need to be validated
Thank you for the comment. I agree with you. I added in the discussion a explaining and clear sentence about that.
Point 2: The work conducted is very thorough, albeit just for one year. Further validation will be required. Especially as there was low disease risk for botrytis and sour rot and high for mildew. Will products work if mildew risk is lower and botrytis higher?
Response 2: Thank you for the comment but as reported in the manuscript the experiments on both cultivars (Nero d’Avola and Inzolia) were conducted in duplicate (in two different sites of vineyards). Although the experiments were not conducted over two seasons they were repeated spatially. For this reason the authors provided ANOVA effects for each target pathogen and post-hoc analysis. In addition, we reported photos of disease symptoms. As regards product performance, pressures of disease infections depend on cultivar (red o white grape berried) and however detected disease parameters, especially on Nero d’Avola, are not so low after all.
Point 3: I also feel the presentation of results in many repetitive tables can be condensed by omitting the ANOVA tables and mentioning the significant trends and interaction in the text (which is already largely done), and by combining individual disease tables into one for easier oversight.
Response 3: Since the experiments were spatially performed two times, the authors believe that ANOVA interactions tables are needed to a more clear understanding of each factor effects on disease parameters. For example, is it not true from these tables reader can be understand that cultivar play an important role on disease infection?... whereas not significant effects are not detected for first and second order interactions and for site factor.
Point 4: I strongly support the need for alternatives in organic and conventional horticulture. I am aware that disease thresholds are important factors to consider in product efficacy and i felt the authors could discuss this more comparing their results with previous work. The cost of replacing sulphur and/or adding additional treatments should also be mentioned. Would the growers be happy to step away from sulphur? with the data shown i would not (yet) because of mildew.
Response 4: Thank you for the comment. However, the authors clearly expressed in different part of manuscript (above results and discussion) that COS-OGA applications always provided lower performances than Cu-S applications. Indeed, these IR applications could be encouraged as alternative also for its largely debated positive aspects.
Point 5: The AMF application onto canopy (?) is unclear. Did the AMF change?
Response 5: See previous detailed response to your comment.

Reviewer 3 Report
The present manuscript sounds good, but I am afraid about the results and discussion were supported only by ANOVA. In my view point the results must be treated by Tukey or other statistical test to check the significative differences among treatment.
Author Response
Response to Reviewer 3 Comments
Point 1: The present manuscript sounds good, but I am afraid about the results and discussion were supported only by ANOVA. In my view point the results must be treated by Tukey or other statistical test to check the significative differences among treatment..
Response 1: Thank you for your positive comment and comprehensive evaluation. As regards statistical data analysis, we at first applied ANOVA to analyse factor and interactions effects and following it we perform a post-hoc analysis. In detail, we have chosen to apply Fisher’s test (among the most used for scientific experiments) to evaluate differences of treatment performances. Obtained ranking of treatment effectiveness obtained with this test is very similar to those obtained with more conservative test as Tukey that you also suggest.

Round 2
Reviewer 2 Report
Earlier comment: Line 106 – the whole sentence needs correcting. As 7 and 26 worked on powdery in grape with the COS-OGA complex. And you want to build on that work and also the AMF work. So tell me more about the work by 7 and 26. And still waiting for the rationale why AMF and not PGRP or any of the others IRs, BCAs mentioned. You also chose three diseases, that probably could do with an introduction and how they affect organic wine production.
Ok, thank you .... We have chosen AMF accordingly to agronomic practice performed in commercial farm where the experiments were carried out. The authors focused also their attention on the three pathogens because they represent the key issues for the Mediterranean area (south Italy). To this regard, it is notherworthy underline the experiments were performed under natural infection pressure.
20 July: my earlier comment really has not been addressed in the manuscript itself. Please append the objectives of this research section accordingly, so the reader understands the why you chose as you did, not just the what
earlier comment: line 370 - four replicates consisting of 12 plants… any buffers between plants? what did the block consist of? 4 rows (one per replicate?)
Thank you, modification done. You are correct..blocks consist of 4 rows. Along single block the treated 12 plants were separated from the subsequent 12 plants differedly treated by an untreated plant and so on. In addition, between rows (blocks) an untreated row was also a buffer row.
please amend the text accordingly. I see you clarified the text somehow, but you did not describe the layout with the buffers as above. please do add. Why would i waste my taime asking? if you do not amend the text?
line 434. why would you spray AMF to the canopy?
line 477: you still don't mention the fungal structures you are observing under the microscope?
Author Response
earlier Comments
Line 106 – As 7 and 26 ………..And still waiting for the rationale why AMF and not PGRP or any of the others IRs, BCAs mentioned. You also chose three diseases, that probably could do with an introduction and how they affect organic wine production.
Ok, thank you for the ………...
Point 1: …..my earlier comment really has not been addressed in the manuscript itself. Please append the objectives of this research section accordingly, so the reader understands the why you chose as you did, not just the what.
Response 1: Ok, you are correct. The authors addressed the reasons supporting their choices (AMF and target papthogens) and added relative sentences in the revised manuscript.
line 370 - four replicates consisting of 12 plants… any buffers between plants? what did the block consist of? 4 rows (one per replicate?)
Thank you, modification done. You are correct..blocks consist of 4 rows. Along single block the treated 12 plants were separated from the subsequent 12 plants differedly treated by an untreated plant and so on. In addition, between rows (blocks) an untreated row was also a buffer row.
Point 2: …..please amend the text accordingly. I see you clarified the text somehow, but you did not describe the layout with the buffers as above. please do add. Why would i waste my taime asking? if you do not amend the text?
Response 2: Ok, you are right but these details are almost always missing in these scientific papers to not overload the manuscript. However, we added the required information in the revised MS
Point 3: ….. line 434. why would you spray AMF to the canopy?
Response 3: We have done these spray treatments according to manufacturer indications reported in the product label
Point 4: ….. line 477: you still don't mention the fungal structures you are observing under the microscope?
Response 4: You are correct. We have added the relative sentences to the observed structures in the revised MS.

Reviewer 3 Report
Dear Author
In think that the new version is ok. The Authors made the corrections and inclusions that improved the quality of the present manuscript.
Author Response
Response to Reviewer 3 Comments
Point 1: In think that the new version is ok. The Authors made the corrections and inclusions that improved the quality of the present manuscript.
Response 1: Thank you for the positive comment. I agree with you.
